# Phenotypic and Genomic Difference among Four *Botryosphaeria* Pathogens in Chinese Hickory Trunk Canker

**DOI:** 10.3390/jof9020204

**Published:** 2023-02-04

**Authors:** Tianling Ma, Yu Zhang, Chenyi Yan, Chuanqing Zhang

**Affiliations:** Department of Plant Pathology, Zhejiang Agriculture and Forest University, Hangzhou 311300, China

**Keywords:** trunk canker, *Botryosphaeria* species, Biolog phenotype microarray, species-specific genes, rapid identification

## Abstract

*Botryosphaeria* species are amongst the most widespread and important canker and dieback pathogens of trees worldwide, with *B. dothidea* as one of the most common *Botryosphaeria* species. However, the information related to the widespread incidence and aggressiveness of *B. dothidea* among various *Botryosphaeria* species causing trunk cankers is still poorly investigated. In this study, the metabolic phenotypic diversity and genomic differences of four Chinese hickory canker-related *Botryosphaeria* pathogens, including *B. dothidea*, *B. qingyuanensis, B. fabicerciana,* and *B. corticis,* were systematically studied to address the competitive fitness of *B. dothidea*. Large-scale screening of physiologic traits using a phenotypic MicroArray/OmniLog system (PMs) found *B. dothidea* has a broader spectrum of nitrogen source and greater tolerance toward osmotic pressure (sodium benzoate) and alkali stress among *Botryosphaeria* species. Moreover, the annotation of *B. dothidea* species-specific genomic information via a comparative genomics analysis found 143 *B. dothidea* species-specific genes that not only provides crucial cues in the prediction of *B. dothidea* species-specific function but also give a basis for the development of a *B. dothidea* molecular identification method. A species-specific primer set Bd_11F/Bd_11R has been designed based on the sequence of *B. dothidea* species-specific gene *jg11* for the accurate identification of *B. dothidea* in disease diagnoses. Overall, this study deepens the understanding in the widespread incidence and aggressiveness of *B. dothidea* among various *Botryosphaeria* species, providing valuable clues to assist in trunk cankers management.

## 1. Introduction

Chinese hickory (*Carya cathayensis* Sarg.), known for the distinctive fragrance and the high nutritional value of its nuts, is an endemic tree species in China [1,2]. Currently, there are about 1,300,000 ha of Chinese hickory cultivated in Zhejiang and Anhui Provinces, which produce an average annual output value of USD 260 million and a processing output value of USD 509 million [3]. However, due to the traditional cultivation methods, monoculturing of single varieties, overfertilization, and excessive application of herbicides, trunk cankers have become the most devastating disease, resulting in severe damage and death to *C. cathayensis* [4]. In the Zhejiang and Anhui Provinces, nearly 90% of orchard trees have been affected by this disease, and this canker disease continues to expand in range, which seriously threatens the sustainability of the Chinese hickory industry [4,5].

*Botryosphaeria* Ces. & De Not. species are the most widespread canker and dieback pathogens of a variety of host woody plants worldwide [6,7], including citrus [8,9], apple [10], and pistachio [11]. The fungal species *Botryosphaeria dothidea* (Moug. ex Fr.) Ces. & De Not. is one of the most common *Botryosphaeria* species on numerous hosts [4,6,12]. Our previous study found that *Botryosphaeria* species is also associated with the trunk cankers in Chinese hickory trees, and *Lasiodiplodia theobromae* (Pat.) Griff. & Maubl., together with four *Botryosphaeria* species, including *B. dothidea*; *Botryosphaeria qingyuanensis* G.Q. Li & S.F. Chen; *Botryosphaeria fabicerciana* (S.F. Chen, D. Pavlic, M.J. Wingf., & X.D. Zhou) A.J.L. Phillips & A. Alves; and *Botryosphaeria cortices* (Demaree & Wilcox) Arx & Muller were identified as Chinese hickory canker-causing agents [5]. Interestingly, among four *Botryosphaeria* species, *B. dothidea* was also considered to be the dominant species causing Chinese hickory cankers due to its highest isolation frequency and strongest virulence, while the other three species were considered “weak” pathogens [5]. However, the reason accounting for the widespread incidence and aggressiveness of *B. dothidea* among various *Botryosphaeria* species causing trunk cankers has not been addressed [5,12,13].

Fungi are physiologically diverse, possessing disparate genomic features and environmental adaptation mechanisms [14]. The information initially encoded in fungal genome is ultimately displayed at the cellular level as functional traits [15,16]. The varieties of functional traits in fungi reflect species-specific life strategies that responsible for their environmental fitness. For instance, regulation of genes involved in nitrogen metabolism plays a role in the ability of fungi to exploit and survive under different environmental situations [17,18,19,20]. Additionally, capability of either osmotic stress tolerance or variant pH adaptation is important for organism overcoming environmental selective pressures [21,22]. Accordingly, large scale analysis from genomic features to physiologic traits tracking microorganism environmental adaptation strategies can be effective to decode the wider prevalence and distribution of dominant species in community.

Traditionally, cellular phenotypes are analyzed one at a time, and efficient measure with adequate scope and sensitivity for full-scale phenotypic analysis was difficult [16,23,24]. Recently, a phenotypic MicroArray/OmniLog system (PMs), with high throughput, has been developed by Biolog Inc. (Hayward, CA, USA). PMs are capable for providing a comprehensive scan of hundreds or thousands physiologic traits and enable the large-scale screening of the metabolic variety that accounts for the phenotypic difference (such as pathogenicity, stress sensitivity) among species [16,23,24]. This system has been applied to analyze the physiological features of many microorganisms, including *Bacillus subtilis* Cohn [25], *Escherichia coli* Castellani & Chalmers [15,16], *Ralstonia solanacearum* (Smith) Yabuuchi [26], *Alternaria alternata* (Fr.) Keissl. [27], and *Botrytis cinerea* Persoon [24]. In this study, large-scale screening of physiologic traits of above four *Botryosphaeria* species that cause Chinese hickory canker were carried out via PM 3 (nitrogen utilization), PM 5 (biosynthetic pathways), PM 9 (osmotic pressure), and PM 10 (ionic strength and pH). *B. dothidea* was found to have a broader spectrum of nitrogen source, greater tolerance toward osmotic pressure (sodium benzoate) and alkali stress among *Botryosphaeria* species. Moreover, the comparative genomics analysis found 143 *B. dothidea* species-specific genes that may responsible for the species-specific functions and not conserved among species. In addition, a species-specific primer set Bd_11F/Bd_11R were designed based on the above comparative genomics analysis for the accurate identification of *B. dothidea* in diseases diagnosis, prevention, and control. Collectively, this study deepens the understanding for the widespread incidence and aggressiveness of *B. dothidea* among various *Botryosphaeria* species will ideally assist in the development of effective measures for trunk cankers management.

## 2. Materials and Methods

### 2.1. Origin and Collection of Botryosphaeria Species

The four *Botryosphaeria* species, including *B. dothidea* strain BDLA16−7, *B. corticis* strain BCTK16−35, *B. fabicerciana* strain BFLG18−2, and *B. qingyuanensis* BQTK16−30, used in phenotypic characterization and genomic analysis were collected from Chinese hickory tree (cultivar of linan) in Linan, Zhejiang province of China. The four stains have been identified via phylogenetic analyses of *ITS*, *β-TUBLIN*, and *EF* gene sequences combined with morphological analyses in our previous study [5], and their raw sequence data and the whole genome sequence data reported are available at Genome Sequence Archive (GSA, https://ngdc.cncb.ac.cn/gsa/), and the Genome Warehouse (GWH, https://ngdc.cncb.ac.cn/gwh), respectively, in National Genomics Data Center, China National Center for Bioinformation (CNCB-NGDC Members and Partners, 2021) [28], under Bioproject PRJCA005744. The *B. dothidea* stains used for the accuracy and efficiency detection of species molecular identification method, including one sequenced strain BDLA16−7 [28], 89 identified *B. dothidea* isolated from diseased Chinese hickory in our previous study [5], and one identified *B. dothidea* stain associated with citrus branch diseases isolated by Xiao et al. [29]. For each strain used in this study, the single-conidium isolates were stored on potato dextrose agar (PDA; 200 g L^−1^ potato boiled for half an hour and strained, 20 g L^−1^ glucose, 12 g L^−1^ agar) slants at 4 °C.

### 2.2. Phenotypic Characterization of Botryosphaeria Species

Large-scale screening of physiologic trait of four *Botryosphaeria* species including *B. dothidea* strain BDLA16−7, *B. corticis* strain BCTK16−35, *B. fabicerciana* strain BFLG18−2, *B. qingyuanensis* BQTK16−30 was conducted using a phenotypic MicroArray/OmniLog system according to the Biolog standard procedure [15,16]. The concentration of the spore suspension of each tested *Botryosphaeria* species was adjusted to 1 × 10^6^ conidia/mL before metabolic plate tests. The conidia suspension of each tested *Botryosphaeria* species was prepared as mentioned in our previous study [5]. Briefly, spores were collected from the 8 to 10 d old colonies grown on PDA plates by pouring 2 to 3 mL of sterile water onto plates, filtering through duplicated gauze, and counting spores with a hemocytometer. Metabolic plates were used to assess the metabolic pathways of nitrogen utilization (PM 3), biosynthetic pathways (PM 5), osmotic pressure (PM 9) and ionic strength and pH (PM 10). To each well of the PM plates, 100 µL of a cell suspension was added with 62% transmittance. The metabolic phenotype test plate was placed in the Biolog system incubator at 28 °C for 96 h. Phenotypic data were recorded every 15 min by the OmniLog 2.4 system capturing digital images of the microarrays and storing turbidity values. Data were then analyzed using Kinetic and Parametric software (Biolog). Phenotypes were estimated according to the area of each well under the kinetic curve of dye formation. The experiment was repeated three times. Heat maps of phenotype analysis was conducted with the software of HemI (Heatmap IIIlustrator, version 1.0).

### 2.3. Comparison of Genomic Differences among Botryosphaeria Species

Orthologous gene clusters analysis was conducted by OrthoFinder v2.5.4 using all of sequences from the four *Botryosphaeria* species with defalt parameters. Those genes coding unclusted proteins and proteins in species-specific clusters were selected as candidate *B. dothidea* species-specific genes. These genes were further search against with genome sequences of other three *Botryosphaeria* species by NCBI-BLASTn, and genes were absent (0 hits) in all the other three *Botryosphaeria* species were retained as *B. dothidea* species-specific genes.

### 2.4. B. dothidea Rapid Identification Method with Species-Specific Primers

To rapidly identification of *B. dothidea,* the species-specific genes with length around 1000 bp (listed in the Appendix A) were chose for species-specific primer design using Primer3 online web service (https://primer3.org/). For each specific primer the following parameters were checked: primer length, primer melting temperature, GC content, GC clamp, primer secondary structures (hairpins, self-dimer, and cross dimer), repeats, runs, 3′ end stability, avoid template secondary structure, optimum annealing temperature, primer pair tm mismatch calculation. In addition, all species-specific primers were further tested for specificity using web-based Primer-BLAST tool (http://www.ncbi.nlm.nih.gov/tools/primer-blast/index.cgi) to avoid the cross reactivity with the other three *Botryosphaeria* species. Then primer sets that have no double-ended match with the other three *Botryosphaeria* species in silico analysis were applied to PCR assays for the screen of the best primers with highest amplification efficiency and specificity.

For the preparation of genomic DNA used in PCR assays, the four *Botryosphaeria* species were incubated on PDA plates at 25 °C in the dark. After mycelia grew to cover nearly two-thirds of the PDA plate surfaces, the cultures were transferred to a mortar and ground with liquid nitrogen. The resultant powder was placed in a 2 mL centrifuge tube and the mycelial DNA from each isolate was extracted using a Genomic DNA Kit (Sangon Biotech Co., Ltd., Shanghai) according to the manufacturer’s instructions.

The PCR was performed in an Applied Biosystems Veriti Thermal Cycler in 50 μL PCR reaction systems comprising 25 μL of 2 × Rapid Taq master mix (Vazyme, P222-01, Nanjing, China), 1 μL of fungal DNA template, 2 μL of 10 μmol· liter^−1^ upstream and downstream primers, and 20 μL of ddH_2_O. PCR reaction conditions included 94 °C for 5 min for denaturation, followed by 35 cycles of 94 °C for 30 s, annealing 57 °C for 30 s, and extension at 72 °C for 45 s, with a final extension at 72 °C for 10 min. Agarose gel electrophoresis (1%) stained with ethidium bromide was then used to verify successful PCR amplification. The experiment was repeated three times.

## 3. Results

### 3.1. Phenotypic Characterization of Four Botryosphaeria Species

The PM 3 plate tests characterizing the abilities of four *Botryosphaeria* species in 95 nitrogen substrates (amino acids) metabolism found *B. dothidea* presented the highest nitrogen utilization ratio (96.8%), followed by *B. fabicerciana* (89.5%), *B. qingyuanensis* (88.4%) and *B. corticis* (86.3%) (Figure 1, Table 1 and Appendix A). In addition, approximately 92 tested compounds were effectively utilized by *B. dothidea*, except D-serine (plate PM 3, well C8), D-valina (plate PM 3, well C9) and D-galactosamine (plate PM 3, well E9) (Figure 1 and Appendix A). In addition, nitrogen sources, including the D-Alanina (plate PM 3, well C3), D-glutamic (plate PM 3, well C6), D-lysine (plate PM 3, well C7), and acetamine (plate PM 3, well E4), were only efficiently metabolized in *B. dothidea* but not the other three *Botryosphaeria* species (Figure 1 and Appendix A). Surprisingly, for 95 biosynthetic pathways tested via PM 5 plate, all three *Botryosphaeria* species showed 100% utilization, while *B. qingyuanensis* demonstrates the inefficiency in L-lysine (plate PM 5, well B6) utilization (Figure 2, Table 1 and Appendix A). Taken together, these results revealed the variety among four *Botryosphaeria* species in the above metabolic and biosynthetic pathways, and *B. dothidea* demonstrated a broader spectrum of nitrogen source among *Botryosphaeria* species.

Using the PM 9 plate, phenotypes of the four *Botryosphaeria* species under various osmotic stresses were tested. As shown in Figure 3, all the four fungi showed active metabolism in utilizing up to 6% potassium chloride (plate PM 9, wells D1 to D4), up to 5% sodium sulfate (plate PM 9, wells D5 to D8), up to 6% sodium, up to 20% ethylene glycol (plate PM 9, wells D9 to D12), up to 6% sodium formate (plate PM 9, wells E1 to E6), up to 12% sodium lactate (plate PM 9, wells F1 to F12), and up to 100 mM sodium nitrite (plate PM 9, wells H1 to H12), while demonstrating significantly attenuated metabolism under the stresses of urea (≥6%, plate PM 9, wells E11 to E12). Interestingly, *B. dothidea* was found to be capable of utilizing up to 50 mM sodium benzoate (plate PM 9, wells G5 to G6), while the other three could only utilize up to 20 mM sodium benzoate (Figure 3 and Appendix A), indicating the possibility that the greater tolerance toward sodium benzoate may account for the widespread incidence and aggressiveness of *B. dothidea*.

The PM 10 plate assessing the tolerance of pH variation found that though all four *Botryosphaeria* species were able to survive under pH variation ranged from 3.5 to 10 (plate PM 10, wells A1-A12), and *B. dothidea,* as well as *B. qingyuanensis,* was more tolerant to strong alkali (Figure 4 and Appendix A). Notably, they were able to maintain active metabolism of multiple nitrogen sources under pH 9.5, for example L-aspartic acid (plate PM 10, well E5), L-glutamic acid (plate PM 10, well E6), L-histidine (plate PM 10, well E9), and L-isoleucine (plate PM 10, well E10), while *B. fabicerciana* and *B. corticis* cannot (Figure 4 and Appendix A), indicating a stronger pH regulation capability in *B. dothidea* and *B. qingyuanensis*.

### 3.2. Comparison of Genomic Differences among Botryosphaeria Species

According to the NCBI-BLASTn results (gene versus genome, cut at ≥ 95% identity, with 0 hits), we revealed 437 (versus *B. corticis*), 440 (versus *B. fabicerciana*) and 185 (versus *B. qingyuanensis*) *B. dothidea* specific protein coding genes (Appendix A). Combine the above unique gene list, *B. dothidea* contained 143 species-specific genes absent in the other three *Botryosphaeria* species (Appendix A). Further analysis of the proteins coded by *B. dothidea* species-specific genes via the KEGG (Kyoto Encyclopedia of Genes and Genomes) pathways (https://www.genome.jp/tools/kofamkoala/) online service found that only 7 proteins were assigned the corresponding KEGG Orthologs (KOs) (Table 2), which may be responsible for the species-specific functions. Additionally, among 7 KEGG annotated proteins, one protein (jg10977) was associated with the calcium permeable stress-gated cation channel, which may have potential roles in the specific osmotic stress tolerance of *B. dothidea* mentioned above.

### 3.3. Design and Application of Primers for B. dothidea Molecular Identification

As showed in Figure 5, the screened primer set Bd_11F (5′-TCCAACGACGAGCAATCC-3′) and Bd_11R (5′-TGTGCCCTGAGGCGGTAT-3′) designed based on the sequence of *B. dothidea* species-specific gene *jg11* was found to give an amplicon of 695 bp for *B. dothidea* strains, including one sequenced strain BDLA16−7 [28], 89 identified *B. dothidea* isolated from diseased Chinese hickory in our previous study [5], and one identified *B. dothidea* stain associated with citrus branch diseases [29], but not for *B. qingyuanensis*, *B. fabicerciana,* and *B. corticis*. Taken together, these results showed that sequence of above strain *B. dothidea* specific genes not only give a predictive information of species-specific function annotation (Table 2) but also provide a basis for the development of *B. dothidea* molecular identification method. Additionally, the species-specific primer set Bd_11F/Bd_11R was able to carry out the accurate identification of *B. dothidea* with high efficiency (Figure 5B).

## 4. Discussion

*Botryosphaeria* species are amongst the most widespread and important canker and dieback pathogens of trees worldwide [6,7], with *B. dothidea* as one of the most common *Botryosphaeria* species [4,5,6,12]. In Chinese hickory trees cankers disease, four Botryosphaeria species, including *B. dothidea*, *B. qingyuanensis*, *B. fabicerciana*, and *B. corticis,* were identified as Chinese hickory canker-causing agents [5]. Consistently, *B. dothidea* was also the dominant species due to its highest isolation frequency and strongest virulence [5]. However, the information related to the widespread incidence and aggressiveness of *B. dothidea* among various *Botryosphaeria* species causing trunk cankers is poorly understood [4,5,12]. In this study, the metabolic phenotypic diversity of four Chinese hickory canker-causing *Botryosphaeria* pathogens was systematically studied to address the competitive fitness of *B. dothidea* via PM 3 (nitrogen utilization), PM 5 (biosynthetic pathways), PM 9 (osmotic pressure), and PM 10 (ionic strength and pH). For nitrogen source metabolization, the highest utilization ratio (96.8%) was found in *B. dothidea*, followed by *B. fabicerciana* (89.5%), *B. qingyuanensis* (88.4%) and *B. corticis* (86.3%) (Figure 1, Table 1 and Appendix A). The efficient utilization of available nitrogen is highly associated with the ability of organism to survive under different environmental situations [30]. For instance, the capability of the methylotrophic yeast *Candida boidinii* Ramirez to adapt a change in the major available nitrogen source from nitrate to methylamine during the host plant aging was crucial for yeast survival on the leaf environment [18,31]. In addition, the atmospheric N_2_ fixing endosymbiosis in *Ustilago maydis* (DC.) Corda, a plant parasitic fungus causing corn smut disease [32], is also a flexible process important for the adaptation of the fungus to survive and grown in media lacking nitrogen [19,33]. Intriguingly, our previous study found that *B. dothidea* was the most frequently isolated species followed by *B. fabicerciana*, *B. qingyuanensis,* and *B. cortices* [5], which is consistent with the above nitrogen source utilization ratio of the four *Botryosphaeria* species. These results indicated a possible positive correlation between nitrogen utilization capability and fungal environmental fitness, and the broader spectrum of nitrogen source utilized by *B. dothidea* may be responsible for its competitive fitness among *Botryosphaeria* species. However, more researches should be performed to verify this hypothesis in the next study.

Osmotic stress has detrimental effects on microbial survival [22,34,35]. The PM 9 plate testing the phenotypes of the four *Botryosphaeria* species under various osmotic stresses found that *B. dothidea* could utilize up to 50 mM sodium benzoate (plate PM 9, wells G5 to G6), while the other three could only utilize up to 20 mM sodium benzoate (Figure 3 and Appendix A). Sodium benzoate has been reported to as useful agent with antimicrobial properties that able to inhibit the growth of many microbes, including *Pseudomonas aeruginosa* (Schroeter) Migula [36], *Penicillium commune* Thom [37], *E. coli*, *Staphylococcus aureus* Rosenbach, *Saccharomyces cerevisiae* Meyen ex E.C. Hansen [38], and *Candida albicans* (Robin) Berkhout [39], indicating the possibility that the greater tolerance toward sodium benzoate caused osmotic stress may account for the environmental fitness of *B. dothidea*. Additionally, the comparative genomics analysis between *B. dothidea* and the other three *Botryosphaeria* species found 143 *B. dothidea* species-specific genes, and further analysis of the above species-specific genes via the KEGG pathways found *jg10977* has a predicated role in the calcium permeable stress-gated cation channel. This channel exists in eukaryotic cells from yeasts to animals and plants, and acts as a primary regulator of the initial responses to osmotic pressure [40,41]. Thus, investigating the potential contribution of *jg10977* in the specific osmotic stress tolerance of *B. dothidea* mentioned above could be a valuable degree to further explore the mechanism involved in the wide prevalence and aggressiveness of *B. dothidea*.

High pH imposes severe stress on the fungal cell, including difficulties in the acquisition of nutrients or reduced availability of essential elements, such as iron or copper [42,43]. Appropriate responses to ambient pH govern fungal virulence during the infection [44]. For example, *Sclerotinia sclerotiorum* (Libert) de Bary can secrete organic acids acidifying the environment and ensure the proper deployment of cell wall–degrading enzymes (CWDEs) and other pathogenicity factors that function at an optimal pH for their activity during the infection [45]. Additionally, the dedicated Pal/Rim signaling pathway that functions in the response to alkaline pH was shown to be essential for infection in a number of fungal pathogens [42,46], such as *C. albicans* [47], *Fusarium oxysporum* Schlechtendahl [48], and *Aspergillus fumigatus* Fresenius [49]. In this study, PM 10 plate test assessing the tolerance of pH variation found that *B. dothidea* and *B. qingyuanensis* were more tolerant to strong alkali among four *Botryosphaeria* species (Figure 4 and Appendix A), indicating a better ability to sense and respond to alkali environment. Significantly, the virulence test in our previous study found the *B. dothidea* was the most virulent, closely followed by *B. qingyuanensis*, then *B. fabicerciana* and *B. cortices* [5]. Consider the close relation between pH adaptation and fungal virulence [42,44,45], we speculated that the variety in pH regulation capability may be responsible for the virulence differentiation among four *Botryosphaeria* species.

In the past, the identification of the *B. dothidea* associated with Chinese hickory trunk cankers mainly relied on morphological identification [5]. In 2019, we developed a quantitative loop-mediated isothermal amplification (q-LAMP) technique to quantitatively monitor *B. dothidea* in environmental samples based on the different sites of the *β-TUBLIN* sequence between *B. dothidea* and other fungi commonly found on Chinese hickory [3]. In 2011, we used the combination of morphological identification and multi-locus phylogenetic analysis of *ITS, β-tublin,* and *EF* gene sequences for the accurate distinction of *B. dothidea* [5]. However, this technique is laborious, expensive, and requires time and knowledge of phylogenetic analysis for identifying the species. Alternatively, the identification of fungal pathogens through a conventional polymerase chain reaction (PCR) using species-specific primers provide an accurate, reproducible, and rapid identification of the species and has been widely used, particularly for economically important plant pathogens [50,51]. In this context, using a comparative genomics analysis, we identified 143 *B. dothidea* species-specific genes that are not conserved among species (Appendix A), and provided a basis for the development of the *B. dothidea* molecular identification method. The screened primer set Bd_11F (5′-TCCAACGACGAGCAATCC-3′) and Bd_11R (5′-TGTGCCCTGAGGCGGTAT-3′) designed based on the sequence of *B. dothidea* species-specific gene *jg11* was found to be used in the accurate identification of *B. dothidea* with efficiency and high efficiency (Figure 5B), which is of significant importance for *B. dothidea*-associated cankers diagnosis, prevention, and control.

Collectively, using the large-scale screening of physiologic traits of four *Botryosphaeria* species associated with Chinese hickory cankers, we found *B. dothidea* has a broader spectrum of nitrogen sources and a greater tolerance toward osmotic pressure (sodium benzoate) and alkali stress among *Botryosphaeria* species. Moreover, the annotation of *B. dothidea* species-specific genomic information via a comparative genomics analysis not only gives a basis for the development of the *B. dothidea* molecular identification method but also provides crucial cues in the prediction of *B. dothidea* species-specific function that account for its high prevalence and wide distribution. However, the speculation of the mechanism involved in the wide prevalence and aggressiveness of *B. dothidea* in this study still require further study to verify.

## Figures and Tables

**Figure 1 jof-09-00204-f001:**
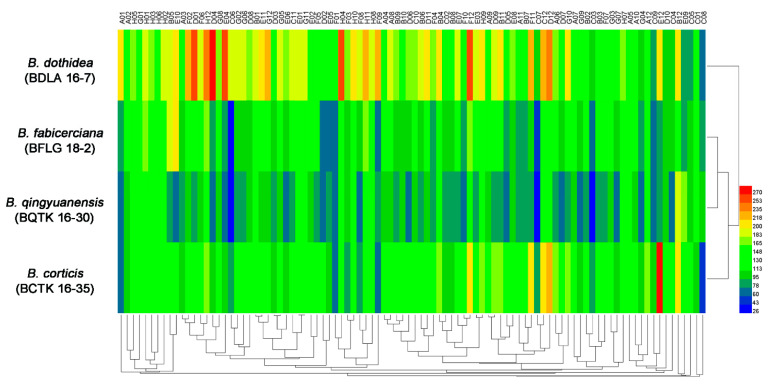
**|** Overview of metabolic phenotypes of four *Botryosphaeria* species, including *B. dothidea* strain BDLA16−7, *B. corticis* strain BCTK16−35, *B. fabicerciana* strain BFLG18−2, and *B. qingyuanensis* strain BQTK16−30, on 95 nitrogen sources tested using the PM 3 plate. Heatmap of maximum area values of 95 nitrogen sources expressed as maximum curve area monitored during 96 h of incubation. The legend of color code from blue to green, and red shades represent low, moderate, and high utilization rate of nitrogen sources, respectively.

**Figure 2 jof-09-00204-f002:**
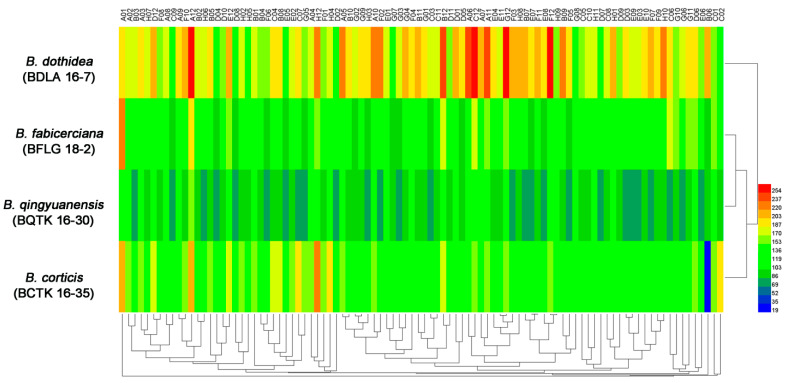
**|** Overview of phenotypes of four *Botryosphaeria* species, including *B. dothidea* strain BDLA16−7, *B. corticis* strain BCTK16−35, *B. fabicerciana* strain BFLG18−2, and *B. qingyuanensis* strain BQTK16-30, on 95 biosynthetic pathways tested using the PM 5 plate. Heatmap of maximum area values of 95 biosynthetic pathways expressed as maximum curve area monitored during 96 h of incubation. The legend of color code from blue to green, and red shades represent low, moderate, and high metabolic rate during the corresponding biosynthetic processes, respectively.

**Figure 3 jof-09-00204-f003:**
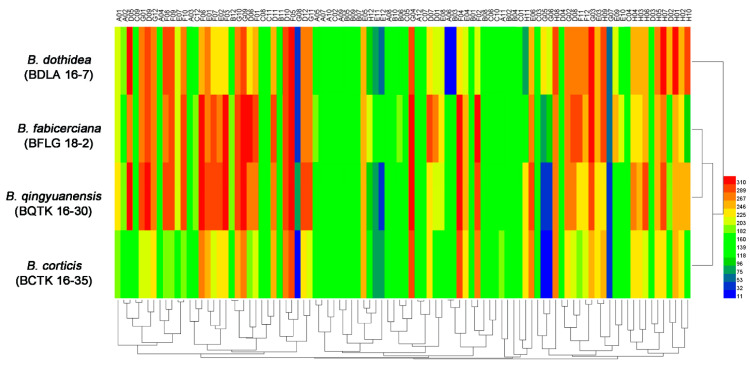
**|** Overview of osmotic stress tolerance of four *Botryosphaeria* species, including *B. dothidea* strain BDLA16−7, *B. corticis* strain BCTK16−35, *B. fabicerciana* strain BFLG18−2, and *B. qingyuanensis* strain BQTK16−30, under 96 osmotic conditions tested using the PM 9 plate. Heatmap of maximum area values of 96 ionic conditions expressed as maximum curve area monitored during 96 h of incubation. The legend of color code from blue to green, and red shades represent low, moderate, and high metabolic rate under the corresponding osmotic stresses, respectively.

**Figure 4 jof-09-00204-f004:**
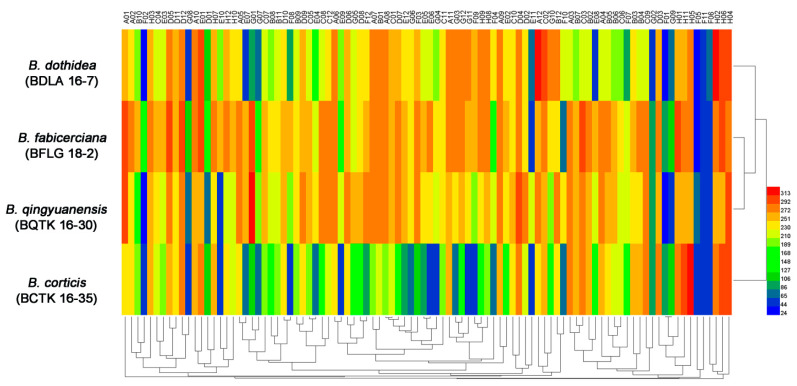
**|** Overview of pH regulation capability of four *Botryosphaeria* species, including *B. dothidea* strain BDLA16−7, *B. corticis* strain BCTK16−35, *B. fabicerciana* strain BFLG18−2, and *B. qingyuanensis* strain BQTK16−30, under 96 pH conditions tested using the PM 10 plate. Heatmap of maximum area values of 96 pH conditions expressed as maximum curve area monitored during 96 h of incubation. The legend of color code from blue to green, and red shades represent low, moderate, and high metabolic rate under the corresponding pH stresses, respectively.

**Figure 5 jof-09-00204-f005:**
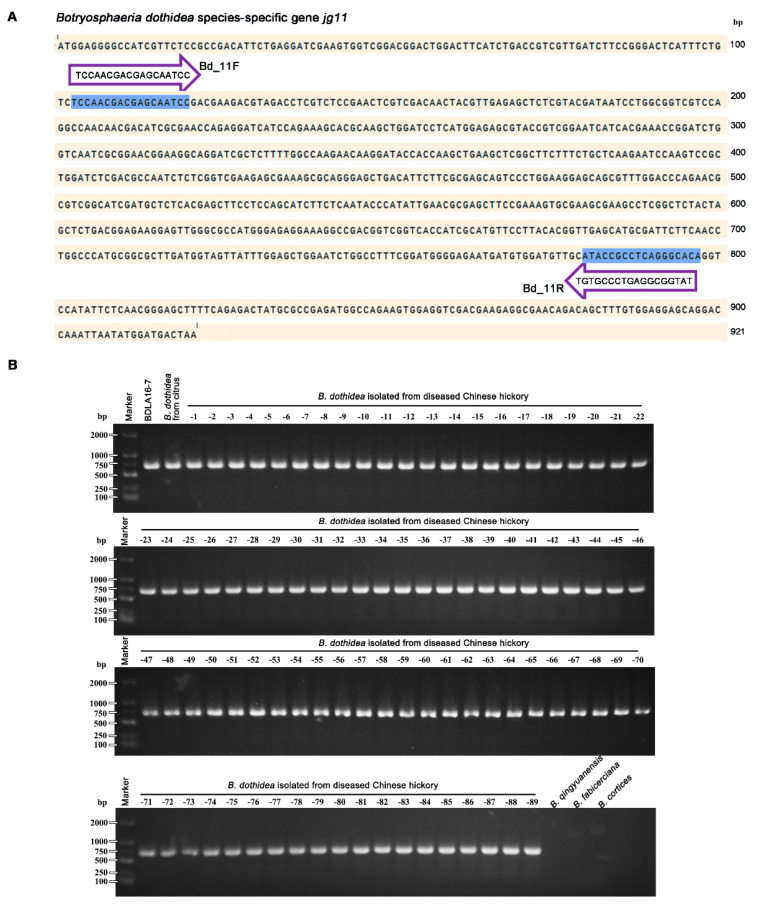
Species−specific primer set Bd_11F/R was able to carry out the accurate identification of *B. dothidea*. (**A**) The species-specific primers Bd_11F and Bd_11R designed based on the species-specific gene *jg11* for molecular identification of *B. dothidea*. The forward and reverse primer sequences were highlighted with shade and arrow for orientation. (**B**) Agarose gel electrophoresis showed the species-specific primer set Bd_11F/R was able to carry out the accurate molecular identification of *B. dothidea*. The primers could give an amplicon of 695 bp for *B. dothidea* strains, including one sequenced strain BDLA16−7, one identified *B. dothidea* stain associated with citrus branch diseases, and 89 identified *B. dothidea* isolated from diseased Chinese hickory, but not for *B. qingyuanensis* (BQTK16−30), *B. fabicerciana* (BFLG18−2), and *B. corticis* (BCTK16−35).

**Table 1 jof-09-00204-t001:** Metabolic ratio of four *Botryosphaeria* spp. on the corresponding Biolog MicroPlates.

Strain	Amino Acid Nitrogen Substrates (%)	Biosynthetic Pathways (%)	Osmotic and Ionic Conditions (%)	pH Condition (%)
*B. dothidea* (BDLA 16−7)	96.8	100	93.8	89.6
*B. fabicerciana* (BFLG 18−2)	89.5	100	93.8	68.8
*B. qingyuanensis* (BQTK 16−30)	88.4	98.9	93.8	89.6
*B. corticis* (BCTK 16−35)	86.3	100	93.8	68.8

Percentages listed in each column is the metabolic ratio of four *Botryosphaeria* spp. on the corresponding Biolog MicroPlates. Metabolic ratio (%) = the number of substrates metabolized by the corresponding *Botryosphaeria* spp. in each kind of Biolog microplate/the total number of the substrates in each kind of Biolog microplate × 100%.

**Table 2 jof-09-00204-t002:** The functional Kyoto Encyclopedia of Genes and Genomes (KEGG) analysis of *B. dothidea* species-specific genes.

Gene Name	KEGG Ortholog(KO)	Thrshld	Score	E-Value	KO Definition
*jg9527*	K23322	48.03	74.2	6.8 × 10^−23^	aspirochlorine biosynthesis cytochrome P450 monooxygenase
*jg3332*	K22993	508.27	561.6	6.3 × 10^−171^	bifunctional Delta-12/omega-3 fatty acid desaturase
*jg10959*	K23504	91.30	266.6	2.9 × 10^−81^	protein SERAC1
*jg10977*	K21989	150.87	173.2	5.8 × 10^−53^	calcium permeable stress-gated cation channel
*g8479*	K01183	118.77	159.1	1.4 × 10^−48^	chitinase
*jg3335*	K01687	579.77	924.5	4 × 10^−280^	dihydroxy-acid dehydratase
*jg3334*	K06867	98.20	108.3	4.4 × 10^−33^	uncharacterized protein

## Data Availability

The data in this study are available in this article.

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
