# Peer review of "Phenotypic and Genomic Difference among Four Botryosphaeria Pathogens in Chinese Hickory Trunk Canker"

_jof, 2023, doi:10.3390/jof9020204_

Round 1

Reviewer 1 Report

 Manuscript "Phenotypic and genomic difference among four Botryosphaeria
pathogens in Chinese hickory trunk canker' is nicely written with clear presentation of ideas. Some of the minor corrections are

typographical errors are visible in pathogen names.

Line no. 93-94 plz combine.

Avoid writing we/she/he in the manuscript.

Line no. 187-193 can be shifted to materials and method. Fig 1, 2, 3 are not clear.

Author Response

Author responses:

We thank the reviewer for the great comments and suggestions, which have helped us to substantially improve the manuscript. Here are the point-by-point responses to the raised comments, and the revised contents in the manuscript was highlighted in red.

Line no. 93-94 plz combine. Avoid writing we/she/he in the manuscript.

-Re: Thanks for the professional reminder. We have fixed the mentioned errors including the font size, typographical errors, the missing spaces, Taxon with identified authors. We also added explanation for the used abbreviation and avoided the first-person narrative perspective in article writing.

Line no. 187-193 can be shifted to materials and method. Fig 1, 2, 3 are not clear.

-Re: Thanks for the kind reminder. We have re-written the results parts according to the professional advice including shifting the contents in Line no.87-89, 129-131, 150-151, 169-172 to introduction part, and shifting the contents in Line no.89-92, 102-103, 187-193 to materials and method part. We also updated the figures and the corresponding legends.

Reviewer 2 Report

Review Report for Journal of Fungi

Manuscript ID: jof-2175283

Title: Phenotypic and genomic difference among four Botryosphaeria pathogens in Chinese hickory trunk canker

Authors: Tianling Ma, Yu Zhang, Chenyi Yan, Chuanqing Zhang

Special Issue: Tree Fungal Disease Problems

 A brief summary outlining the aim of the paper and its main contributions:

The nicely done study with sound results on Botryosphaeria species, which are are important canker and dieback pathogens of trees. In the presented study, the metabolic phenotypic diversity and genomic differences of four Chinese hickory canker related Botryosphaeria pathogens were investigated. Large-scale screening of physiologic traits using a phenotypic MicroArray/OmniLog system (PMs) was performed for B. dothidea, B. qingyuanensis, B. fabicerciana, and B. cortices. B. dothidea was found to be the most capable among the four species with a broader spectrum of nitrogen source, greater tolerance toward osmotic pressure and alkali stress. Based on the conducted comparative genomics analysis, B. dothidea species-specific genes were found and were used to design a B. dothidea species-specific primer set. The retrieved results of this study are maybe a valuable clue to assist in trunk cankers management.

General concept comments

areas of strength:

·         The authors provided very nice figure demonstrating an overview of metabolic phenotypes of studied Botryosphaeria species

·         Very good results in the conducted comparative genomics analysis.

·         The designed species-specific primer set will be very useful in in identifying B. dothidea in infected tissue.

areas of weakness:

·         The manuscript is not following the instructions for authors of Research Manuscript Sections by changing the manuscript order: Introduction, Materials and Methods, Results, Discussion (may be combined with Results) and optional Conclusions.
In the reviewed manuscript the introduction is followed directly by the results Line 84-85. Very often parts of the introduction were inserted in the results and in the discussion

·         In the introduction, the aims of the study should be formulated more clearly.

·         The description of Material and methods and results should be more concise

Specific comments

·         Please add the author, when a specific Taxon is first mentioned.

·         Please use a space before the citation brackets

·         Please pay attention to uniform font size throughout the manuscript

Specific comments referring to line numbers please find in the attached pdf, e.g.:

Line 18: “genomics analysis” wrong font size

Line 30 and subsequent throughout the whole manuscript: A space is missing between the word and the following citation bracket. Please check throughout the whole manuscript and insert a space.

Line 39: author is missing: Please add the author, when the a specific Taxon is first mentioned

Line 41: author is missing: Please add the author, when the a specific Taxon is first mentioned

Line 44-45: author is missing: Please add the author, when the a specific Taxon is first mentioned

Line 70-71: authors are missing: Please add the author, when the a specific Taxon is first mentioned

Author Response

Author responses:

We thank the reviewer for the great comments and suggestions, which have helped us to substantially improve the manuscript. We have updated the manuscript order according to the instructions for authors of Research Manuscript Sections. The following are the point-by-point responses to the raised comments, and the revised contents in the manuscript was highlighted in red.

Specific comments

  • Please add the author, when a specific Taxon is first mentioned.

-Re: Thanks for the professional reminder. We have fixed the mentioned errors including the font size, typographical errors, the missing spaces before the citation brackets, taxon with identified authors, reference format. We also added explanation for the used abbreviation and avoided the first-person narrative perspective in article writing.

  • Please use a space before the citation brackets

-Re: Thanks for the professional reminder. We have fixed that.

  • Please pay attention to uniform font size throughout the manuscript

 -Re: Thanks for the professional reminder. We have fixed that.